# An Efficient Chromatin Immunoprecipitation Protocol for the Analysis of Histone Modification Distributions in the Brown Alga *Ectocarpus*

**DOI:** 10.3390/mps5030036

**Published:** 2022-04-25

**Authors:** Simon Bourdareau, Olivier Godfroy, Josselin Gueno, Delphine Scornet, Susana M. Coelho, Leila Tirichine, J. Mark Cock

**Affiliations:** 1Algal Genetics Group, Integrative Biology of Marine Models Laboratory, CNRS, Sorbonne Université, Station Biologique de Roscoff, CS 90074, F-29688 Roscoff, France; sbourdareau@stowers.org (S.B.); godfroy@sb-roscoff.fr (O.G.); josselingueno@gmail.com (J.G.); scornet@sb-roscoff.fr (D.S.); susana.coelho@tuebingen.mpg.de (S.M.C.); 2Nantes Université, CNRS, US2B, UMR 6286, F-44000 Nantes, France

**Keywords:** brown algae, chromatin, chromatin immunoprecipitation, ChIP-seq, *Ectocarpus*, Phaeophyceae

## Abstract

The brown algae are an important but understudied group of multicellular marine organisms. A number of genetic and genomic tools have been developed for the model brown alga *Ectocarpus*; this includes, most recently, chromatin immunoprecipitation methodology, which allows genome-wide detection and analysis of histone post-translational modifications. Post-translational modifications of histone molecules have been shown to play an important role in gene regulation in organisms from other major eukaryotic lineages, and this methodology will therefore be a very useful tool to investigate genome function in the brown algae. This article provides a detailed, step-by-step description of the *Ectocarpus* ChIP protocol, which effectively addresses the difficult problem of efficiently extracting chromatin from cells protected by a highly resistant cell wall. The protocol described here will be an essential tool for the future application of chromatin analysis methodologies in brown algal research.

## 1. Introduction

Brown algae (Phaeophyceae) are a group of almost exclusively marine photosynthetic eukaryotes. These seaweeds are found along coastlines worldwide, where they have important ecological roles as primary producers and as habitats for a broad range of other species [1]. In recent years, several brown algal species have emerged as important aquaculture crops as a consequence of their capacity to rapidly produce biomass under sustainable cultivation conditions that do not require the use of arable terrestrial land or freshwater resources [2,3]. All brown algae are multicellular, and some species, such as the kelps, exhibit considerable developmental complexity. As members of the Stramenopile supergroup, these seaweeds are very distantly related to other complex multicellular groups, such as animals, land plants, and fungi [4]. Brown algae therefore acquired complex multicellularity independent of their animal and land plant lineages and, consequently, are of interest in order to understand this important evolutionary transition [5].

The small, filamentous alga *Ectocarpus* has been established as a genetic model organism for the brown algae [4,6], and a range of genetic and genomic tools are available. These include the complete genome sequence of *Ectocarpus* strain Ec32, which is publicly available at http://bioinformatics.psb.ugent.be/orcae/overview/EctsiV2 (accessed on 21 April 2022) [7,8], and both forward [9] and reverse genetic methodologies, the latter using a CRISPR-Cas9-based approach [10]. *Ectocarpus* has been used as a model to investigate multiple aspects of brown algal biology, including life cycle regulation [11], molecular mechanisms underlying developmental processes [9,12,13], sex determination [14], and genetic responses to stress [15]. Transcriptomic approaches have commonly been used to characterize gene expression in relation to these various biological processes [9,11,13,15,16]. These analyses have provided information about changes in patterns of gene expression; however, the underlying mechanisms that regulate gene expression in *Ectocarpus* are still poorly understood. To address this knowledge gap, methods are currently being developed to identify and characterize modifications to *Ectocarpus* chromatin associated with the observed changes in gene expression.

Chromatin consists of genomic DNA together with other associated molecules, predominantly proteins. Specific modifications of the core components of chromatin (DNA plus the associated histone molecules that form the nucleosomes) play an important role in the control of gene expression [17]. These modifications can include methylation of the DNA sequence and various post-translational modifications (PTMs, e.g., acetylation, methylation, ubiquitination) of histone proteins [17]. DNA methylation has not been detected in *Ectocarpus* [7], although low levels have been detected in another brown algal species, *Saccharina japonica* [18]. For *Ectocarpus*, therefore, efforts have focused on the development of methodology to characterize histone PTMs.

Chromatin immunoprecipitation (ChIP) has become a widely used method to study the proteins associated with genomic DNA. Combined with high-throughput sequencing, ChIP allows the analysis of the genome-wide distribution of specific histone modifications and the determination of genome-wide binding patterns for transcription factors. Within the Stramenopiles, ChIP protocols have been developed for diatom [19] and oomycete [20] models. More recently, the ChIP method has been adapted for the brown algal model *Ectocarpus*, and has been used to analyze changes in histone PTM patterns associated with alternation between the sporophyte and gametophyte generations during the progression of this species’ haploid–diploid life cycle [11]. This study identified specific histone PTMs associated with transcription start sites and gene bodies of active genes, and with transposons, providing evidence that at least some PTMs play similar roles in brown algae to those described in animals and/or land plants. Surprising differences compared to animals and land plants included the absence of polycomb complexes (and the corresponding histone PTMs) in brown algae, and an unusual distribution of dimethylated lysine 79 of histone H3, which was detected over regions of the genome that often included several contiguous genes [21]. An unusual structural feature of the *Ectocarpus* genome is the common occurrence of closely spaced, divergently transcribed gene pairs [7]. ChIP analysis showed that these gene pairs share a common nucleosome-depleted region, and exhibit shared histone modification peaks [21]. More recently, the ChIP method has been used to compare genome-wide patterns of histone PTMs in male and female strains of *Ectocarpus* [22]. This study allowed the identification of changes in histone PTM patterns at sex-biased genes that were correlated with sex. Interestingly, the pattern of histone PTMs along the sex chromosomes was different to that observed for the autosomes, and this difference appeared to be related to the presence of more evolutionarily young genes on the sex chromosomes [22].

This article provides a detailed, step-by-step description of the ChIP protocol reported by Bourdareau et al. [21,23]. Isolation of complex structures, such as chromatin, from brown algal cells is complicated by the presence of a complex cell wall that both renders the cells highly resistant to cell breakage and represents a source of contaminating biomolecules. The protocol described here employs an optimized cell lysis method that efficiently breaks open the filament cells of the algal thallus, leading to the release of the nuclei without unduly damaging the chromatin. The individual steps of this extraction part of the procedure are described in particular detail, with critical steps highlighted and explained. The protocol described here will be an essential tool for the future application of chromatin analysis methodologies in brown algal research.

## 2. Experimental Design

Here we present an efficient ChIP protocol for the brown alga *Ectocarpus*. This protocol includes the following steps: an optimized crosslinking and nuclei isolation step; an efficient chromatin sonication method, which fragments chromatin to a target size of 300 bp and, finally, immunoprecipitation, reverse crosslinking, and DNA extraction steps that are common to protocols developed for other model species.

The general outline of the experiment is as follows (Figure 1):

### 2.1. Materials and Reagents

Sporophytes and gametophytes of *Ectocarpus* male strain Ec32 (reference CCAP 1310/4 in the Culture Collection of Algae and Protozoa, Oban, UK)Sterile, filtered natural seawaterProvasoli enrichment solution140 mm Petri dishes (Corning, catalogue number: CLS430597)Parafilm (Sigma-Aldrich, Darmstadt, Germany, catalogue number P7793-1EA)Cell scraper (optional)Large-size cell strainer (for example, a 12.5 cm Finlandek permanent coffee filter) (optional, for tissue harvesting after culture)Dissection forceps1 L Erlenmeyer flask37% FormaldehydePBS pH 7.5Sterile Miracloth (Millipore, Watford, UK, ref. 475855)50 mL plastic Falcon tubes (Cellstar^®^, Bishop Stortford, UK, ref. 227 261)Refrigerated centrifuge for 50 mL plastic Falcon tubes (Eppendorf 5804R)Liquid nitrogenPestle and mortarPre-cooled spatula to transfer ground tissue from the mortar7 mL Tenbroeck potter (Kontes, Mainz, Germany ref. 885000-0007)15 mL plastic Falcon tubes (Cellstar^®^, Bishop Stortford, UK, ref. 188 271)1.5 mL and 2 mL microtubesRefrigerated centrifuge for microtubes (Eppendorf 5427 R; maximum RCF 14,000× *g*)DAPI stock solution (2 mg/mL)Microscope slidesMicroscope coverslips (Knittel ref. MS0009)Covaris^®^ microTUBEs (Covaris, Brighton, UK, AFA Fiber Pre-Slit Snap-Cap 6 mm × 16 mm, ref. 520045)SmartLadder and SmartLadder SF DNA size markers (Eurogentech, Seraing, Belgium, MW-1700-02 and MW-1800-02)Normal rabbit IgG naïve antibody (CST 2729)ChIP-grade antibodies against histone modifications of interestDynaBeads protein A (Invitrogen ref. 10002D)DynaBeads protein G (Invitrogen ref. 10004D)1.5 mL and 2 mL Eppendorf^®^ DNA LoBind microtubes (Cat. N° 022431021 and 022431048)Magnetic microtube rack (Invitrogen, Carlsbad, CA, USA, ref. 12321D)Glycine5 M NaCl stock solution0.5 M EDTA pH 8 stock solution1 M Tris-HCl pH 6.5 stock solutionProteinase K 20 mg/mL (Invitrogen, Carlsbad, CA, USA, ref. AM2546)RNAse A, DNase, and protease-free 10 mg/mL (Thermo Fisher Scientific ref. EN0531)Pre-mixed phenol:chloroform:isoamyl alcohol (25:24:1)100% Ethanol70% (*v*/*v*) Ethanol3 M Sodium acetate pH 5.2Glycogen (Thermo Fisher Scientific, Loughborough, UK, ref. R0561)MilliQ waterHigh Sensitivity DNA (Agilent, Santa Clara, CA, USA, ref. 5076-4626)Qubit™ 1X High Sensitivity dsDNA (Invitrogen™ ref. Q33230)

### 2.2. Equipment

Thermostatically controlled, illuminated growth cabinet (POL-EKO-APARATURE, Wodzisław Śląski, Poland, model: KK 240 FIT P) or growth roomThermomixer (Eppendorf, Saint-Quentin-Fallavier, France)Fluorescence microscope (Olympus, Southend-on-Sea, UK, model: CKX41)Covaris^®^ M220 Focused-ultrasonicatorAgarose gel electrophoresis systemBioanalyzer 2100 (Agilent, Santa Clara, CA, USA)Qubit 4.0 (Life Technologies, Paisley, UK)

## 3. Procedure

### 3.1. Growth and Harvesting of Ectocarpus Cultures (Required Time: 2 Weeks)

Inoculate 75 mL aliquots of Provasoli’s enriched natural seawater in transparent 140 mm diameter Petri dishes with the appropriate *Ectocarpus* strain, seal with Parafilm, and grow at 13 °C with a 12 h day/12 h night cycle and 20 µmol photons.m^−2^.s^−1^ irradiance [24]. Culture at least 1200 individual sporophytes or gametophytes at a density of six individuals per Petri dish.After culture, carefully transfer the algae with dissection forceps into a sterile Erlenmeyer flask containing 400 mL of sterile seawater. Two weeks of culture should produce approximately 4–5 g (FW) of tissue.


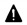
**CRITICAL STEP** For some strains it may be necessary to grow more individuals per Petri dish and harvest at an earlier stage of development before cultures become fertile and produce the next generation of the life cycle. With some sporophyte strains it may be necessary to use a cell scraper to dislodge thalli from the bottom of the Petri dish. This material can be collected by filtering the seawater medium and then transferring the algal material with dissection forceps to the 400 mL of sterile seawater.

### 3.2. Crosslinking (Required Time: 2 h)

3.Crosslink the algal material by adding 400 mL of 2x crosslinking buffer to the 400 mL of algal material in seawater and incubating for exactly 5 min at room temperature under a chemical hood. Mix gently during the crosslinking.


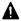
**CRITICAL STEP** Do not incubate for more than 5 min in the 2x crosslinking buffer as this could lead to excessive crosslinking, which may interfere with the chromatin extraction and immunoprecipitation steps. Note that formaldehyde is toxic if inhaled, ingested, or absorbed through skin.

4.Filter the tissue rapidly (15 s maximum) through a sterile piece of Miracloth to eliminate the formaldehyde. Transfer the tissue to a new 50 mL tube containing 50 mL of PBS quenching buffer for 5 min at room temperature. Mix gently during quenching.5.Centrifuge at 3215× *g* 4 °C in an Eppendorf 5804R for 5 min for gametophytes or for 10 min for sporophytes.6.Eliminate the buffer and resuspend in 50 mL of 1x phosphate-buffered saline (PBS) to wash the tissue. Centrifuge at 3215× *g* 4 °C in an Eppendorf 5804R for 5 min for gametophytes, or for 10 min for sporophytes.7.Remove as much of the supernatant as possible by pipetting. You can invert the tube onto a piece of Miracloth placed on absorbent paper. Wrap approximately 1 g batches of tissue in aluminium foil, note the mass of tissue in each batch as this will be used to calculate the volume of nuclei isolation buffer to be added in the following steps. Freeze the crosslinked tissue rapidly in liquid nitrogen.


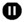
**PAUSE STEP** The crosslinked tissue can be stored at −80 °C at this stage, but it should be used within the next few days if possible, and should not be stored for more than 1 month.


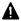
**CRITICAL STEP** The filamentous structure of the material may be partly dissociated after fixation, making it difficult to recover the pellet. If this is the case, scoop the pellet out of the Falcon tube with a spatula.

### 3.3. Isolation of Semipure Nuclei (Required Time: 2 h)

8.Grind about 1 g of crosslinked tissue to an ultra-fine powder under liquid nitrogen using a pre-chilled mortar and pestle. Ensure that samples do not thaw during grinding.


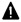
**CRITICAL STEP** It is important that the tissue is very thoroughly ground at this stage. To avoid cross-contamination use different mortars and pestles for different experimental conditions.

9.Transfer the powder to a 15 mL tube containing pre-chilled nuclei isolation buffer with Triton X-100, β-mercaptoethanol, and protease inhibitor cocktail (approximatively 5 mL of buffer for 1 g of tissue). Resuspend well by pipetting up and down.10.Transfer the extract into a 7 mL Tenbroeck potter. Grind 10 times slowly on ice, making hemicircular movements of the potter in the tube when you insert and remove it. To avoid cross-contamination, do not use the same Tenbroeck potter for the different experimental samples.


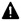
**CRITICAL STEP** To effectively break open the *Ectocarpus* cells, the pestle of the Tenbroeck potter should fit tightly into the cylinder, and it should be quite difficult to move the pestle down the cylinder while making the hemicircular movements. When testing a Tenbroeck potter that has not been used previously, it is important to verify cell lysis using DAPI staining (see below). Grinding is less efficient if too much tissue is extracted; do not exceed 1.5 g.

11.Incubate on ice for 20 min. Resuspend every 5 min.12.Pre-wet two layers of Miracloth by pipetting 500 µL of nuclei isolation buffer with Triton X-100, β-mercaptoethanol, and protease inhibitor cocktail, and then filter the extract through the Miracloth into a 15 mL conical tube on ice. The two layers of Miracloth should be rotated at 90° to each other. Squeeze the Miracloth well to remove all liquid. This step is necessary to remove large debris.13.Aliquot the filtered extract into several 2 mL microtubes and centrifuge for 20 min at 3000× *g* and 4 °C.14.Remove the supernatant and gently resuspend and combine the pellets from the same sample in 1 mL of nuclei isolation buffer with Triton X-100, β-mercaptoethanol, and protease inhibitor cocktail. Centrifuge the extract for 20 min at 3000× *g* and 4 °C. Repeat the wash with 1 mL of nuclei isolation buffer with Triton X-100, β-mercaptoethanol, and protease inhibitor cocktail, then centrifuge the extract for 20 min at 3000× *g* and 4 °C. The pellet should be a pale yellow or whitish colour at this stage, and the supernatant pale brown (Figure 2).

15.Remove the supernatant and gently resuspend the pellets in 1 mL of nuclei isolation buffer with β-mercaptoethanol and protease inhibitor cocktail but without Triton X-100.16.**OPTIONAL STEP** Verify the release of the nuclei using microscopy. Dilute 1 µL of stock solution of 2 mg/mL DAPI in water (stored at −20 °C) in 100 µL of nuclei isolation buffer without Triton X-100, and then add 2 µL of this 10x preparation to 20 µL of extract. Incubate at room temperature for 10 min, then place between a slide and coverslip and visualize under a fluorescence microscope (Figure 3).


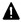
**CRITICAL STEP** DAPI is toxic and a mutagen. Wear gloves while working with DAPI.

### 3.4. Lysis of Nuclei and Sonication (Required Time: 3 h)

17.Transfer to a new 2 mL microtube and centrifuge for 20 min at 3000× *g* and 4 °C. Prepare the nuclei lysis buffer and ChIP dilution buffer at this stage and cool to 4 °C.18.Remove the supernatant and resuspend the pellet in 200 µL to 1 mL (depending on the quantity of starting tissue) of cold nuclei lysis buffer. For example, use 1 mL of nuclei lysis buffer for 1 g of tissue.19.Keep a 5 µL aliquot of the chromatin extract to run on an agarose gel (see below) in order to compare with the chromatin extract after sonication.


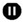
**PAUSE STEP** If necessary, the chromatin extract can be frozen at −80 °C overnight at this stage, but it is better to continue directly to the sonication step if possible.

20.Pipette 130 µL aliquots of the chromatin extract into new, clean Covaris^®^ microTUBEs (AFA Fiber Pre-Slit Snap-Cap 6 mm × 16 mm). It is important to add exactly 130 µL to each microTUBE and to avoid foaming to ensure complete fragmentation.21.Sonicate the chromatin extract in a Covaris^®^ M220 Focused-ultrasonicator™ using the following parameters: duty 25%, peak power 75, cycles/burst 200, time duration 900 s, set point temperature 6 °C (range between 4 °C and 7 °C). Note that sonication could alternatively be carried out with another sonicator, such as the Diagenode Bioruptor Pico, with minor optimization.22.Combine the contents of the microTUBEs after sonication and centrifuge for 5 min at 14,000× *g* and 4 °C to pellet the debris (Figure 2). Transfer the supernatant to a new 1.5 mL microtube. Keep 5 µL of this sample to run on a 0.8% agarose gel (along with the unsonicated sample taken earlier) to visualize the DNA fragments after sonication. After gel electrophoresis, a smear of sonicated chromatin should be observed between 100 bp and 1000 bp (Figure 4). Note that the samples are crosslinked chromatin at this stage (i.e., not naked DNA), so precise estimation of fragment size is not possible; however, gel electrophoresis provides an approximate estimation of the efficiency of fragmentation.


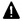
**CRITICAL STEP** Note that the centrifugation step is essential to remove impurities that would otherwise interfere with and contaminate the immunoprecipitation step.

23.Measure the volume of supernatant and dilute tenfold with ChIP dilution buffer. For each sample, keep 50 μL of diluted, sonicated chromatin as an input control. The chromatin extract can be used directly to set up immunoprecipitations, or stored at –20 °C or −80 °C. Note that ChIP dilution buffer is also required to prepare the DynaBeads (Section 3.5). Store an aliquot of ChIP dilution buffer at 4 °C overnight for this.


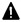
**CRITICAL STEP** Note that it is important to dilute from 1% to 0.1% SDS by adding ChIP dilution buffer because a high concentration of SDS can interfere with epitope–antibody interactions.

### 3.5. Immunoprecipitation (Required Time: 24 h)

24.Prepare 500 μL aliquots of diluted chromatin (equivalent to 100 mg of starting tissue) in 1.5 mL Eppendorf DNA LoBind^®^ microtubes. Add the recommended volume of your antibody. The volume is 5 µL for most antibodies, but optimal volumes may vary and should be determined empirically. As a negative control, add a naïve antibody, such as normal rabbit IgG CST N° 2729. As a mock control, carry out an immunoprecipitation without any antibody. Co-incubate the sonicated chromatin and antibody overnight at 4 °C on a rotating wheel (10 rpm).25.The next day, prepare low-salt wash buffer, high-salt wash buffer, LiCl wash buffer, TE buffer, and elution buffer. Place the first four buffers on ice to cool and pre-heat the elution buffer to 65 °C.26.Prior to use, resuspend DynaBeads protein A and DynaBeads protein G by vortexing. Pre-equilibrate the DynaBeads by pipetting 50 µL of each type of bead into separate, new microtubes, place in the magnetic rack, discard the supernatant and add 50 µL of ChIP dilution buffer, mix well by pipetting up and down. Repeat this wash twice and resuspend each lot of beads in 50 µL of ChIP dilution buffer.27.Mix DynaBeads protein A and DynaBeads protein G in an equal volume (ratio 1:1). There should be 100 µL of beads for each immunoprecipitation.28.Add 100 µL of pre-equilibrated beads to each 1.5 mL Eppendorf^®^ DNA LoBind microtube containing chromatin extract plus antibody (i.e., to all the tubes from step 24). Mix for 2 h at 4 °C with gentle rotation (on a rotating wheel at 10 rpm).29.Separate the supernatant and DynaBeads in a magnetic rack.30.Carry out all washing steps in a cold room, if possible, otherwise keep the samples on ice. Collect immune complexes by placing the tubes in the magnetic rack for 2 min. Remove the supernatant.31.Wash the DynaBeads–IgG–antigen–chromatin complexes twice with each of the following four buffers: low-salt wash buffer, high-salt wash buffer, LiCl wash buffer, and TE Buffer. For each buffer, carry out a rapid first wash, replacing the tube immediately in the magnetic rack, and then a 5 min second wash with the beads in suspension. Use 1 mL of each buffer and wash at 4 °C using only brief, gentle hand inversions to resuspend the beads. To recover the beads, place the tubes in the magnetic rack for 2 min. During this time, invert the rack once to recover any beads in the tube cover. After the 2 min, carefully remove the supernatant and add the next buffer.32.Elute the immune complexes by adding 250 µL of elution buffer (made fresh at step 25 and pre-warmed to 65 °C) to the washed beads. Vortex and incubate in a Eppendorf Thermomixer at 65 °C for 20 min at 1000 rpm. Place the tubes in the magnetic rack and carefully transfer the supernatant fraction (eluate) to another Eppendorf^®^ DNA LoBind tube. Repeat the elution with an additional 250 µL of elution buffer, and combine the two eluates. At the same time, add 450 µL of elution buffer to the 50 µL input control (positive control).

### 3.6. Reverse Crosslinking and RNA/Protein Digestion (Required Time: 18 h)

33.Add 20 µL 5 M NaCl to each tube and incubate at 65 °C for at least 12 h, or overnight, to reverse the crosslinking.34.Add 10 µL of 0.5 M EDTA (pH 8), 20 µL of 1 M Tris-HCl (pH 6.5), 1 µL of 20 mg.mL^−1^ proteinase K, and 1 µL of 10 mg.mL^−1^ RNAse A to the eluate and incubate for 1.5 h at 45 °C.

### 3.7. DNA Extraction and Precipitation (Required Time: 2 h)

35.Add an equal volume (550 µL) of phenol:chloroform:isoamyl alcohol (25:24:1) and vortex briefly.36.Centrifuge the samples in a microcentrifuge at 13,800× *g* for 15 min at 4 °C. Transfer the supernatant to a new 2 mL Eppendorf^®^ DNA LoBind microtube.37.To each tube, add 1.25 mL of 100% ethanol, 50 µL of 3 M sodium acetate (pH 5.2), and 4 µL glycogen (20 mg.mL^−1^). Incubate for 1 h, or overnight, at −80 °C to precipitate the DNA.38.Centrifuge each sample at 13,800× *g* for 15 min at 4 °C.39.Discard the supernatant, wash the pellet with 500 µL of 70% (*v*/*v*) ethanol, centrifuge again at 13,800× *g* for 10 min at 4 °C.40.Discard the supernatant and dry the pellet briefly at room temperature.


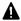
**CRITICAL STEP** Take care not to over-dry the glycogen-containing pellets as they may become insoluble.

41.Resuspend the pellets in 30 µL of DEPC water, store the DNA at −80 °C, and use within 3 months. Traces of phenol may be detected at this stage, but they do not interfere with library construction.

### 3.8. DNA Analysis (Required Time: 2 h)

42.DNA concentration and size range can be analyzed using a Bioanalyzer and a High Sensitivity DNA Chip (Agilent; Figure 5). In addition, DNA concentration should be assessed using a Qubit fluorometer and a dsDNA HS Assay kit.

## 4. Expected Results

This protocol yields approximately 20–25 μg of chromatin per gram of tissue. DNA fragment size should be between 150 bp and 1000 bp (Figure 2). To obtain strong enrichment, we recommend using between 2 μg and 5 μg of antibody and approximately 1.5 μg of chromatin per immunoprecipitation. Ideally, calibration curves should be carried out for each antibody to optimize the efficiency of immunoprecipitation and to minimize background noise. The number of immunoprecipitations required for each antibody should be determined empirically. Table 1 indicates approximate quantities of immunoprecipitated chromatin to be expected when using antibodies against different histone PTMs. At least 5 ng (we recommend 20 ng) of DNA is necessary to prepare sequencing libraries for Illumina platforms using the TruSeq ChIP Library Preparation kit (Illumina). Often, DNA immunoprecipitated using a naïve IgG is not amplifiable, as less than 1 ng can be collected by pooling six immunoprecipitations. For the test samples, we typically pool 4–5 samples per lane on an Illumina HiSeq 4000, aiming for 40–90 million reads per sample in order to obtain strong signals. Inputs should be sequenced more deeply, aiming for 100–110 million reads to accurately model the background and limit detection bias.

## 5. Reagent Setup

### Solutions

2x Crosslinking buffer (2% formaldehyde in natural seawater);PBS quenching buffer (1x PBS:2M glycine in a 4:1 ratio, the final concentration of glycine is 400 mM, PBS is at pH 7.5);Protease inhibitor cocktail (one Roche EDTA-free cOmplete ULTRA tablet dissolved in 1 mL MilliQ water—can be prepared in advance and frozen at −20 °C);Nuclei isolation buffer (125 mM sorbitol, 20 mM potassium citrate, 30 mM MgCl_2_, 5 mM EDTA, 55 mM HEPES pH 7.5; just before use, add 0.1% Triton X-100, 5 mM β-mercaptoethanol, and 2% protease inhibitor cocktail—note that nuclei isolation buffer can be stored for several months at 4 °C);Nuclei isolation buffer without Triton X-100 (125 mM sorbitol, 20 mM potassium citrate, 30 mM MgCl_2_, 5 mM EDTA, 55 mM HEPES pH 7.5; just before use, add 5 mM β-mercaptoethanol and 2% protease inhibitor cocktail);Nuclei lysis buffer (10 mM EDTA, 1% SDS, 50 mM Tris-HCl pH 8, 10% protease inhibitor cocktail—can be prepared in advance and kept at 4 °C);ChIP dilution buffer (1% Triton X-100, 1.2 mM EDTA, 167 mM NaCl, 16.7 mM Tris-HCl pH 8.0—note that ChIP dilution buffer can be stored for several months at 4 °C);Low-salt wash buffer (150 mM NaCl, 0.1% SDS, 1% Triton X-100, 2 mM EDTA, 20 mM Tris-HCl pH 8.0);High-salt wash buffer (500 mM NaCl, 0.1% SDS, 1% Triton X-100, 2 mM EDTA, 20 mM Tris-HCl pH 8.0);LiCl wash buffer (0.25 M LiCl, 1% NP40, 1% sodium deoxycholate, 1 mM EDTA, 10 mM Tris-HCl pH 8.0);TE buffer (1 mM EDTA, 10 mM Tris-HCl pH 8.0);Elution buffer (1% SDS, 0.1 M NaHCO_3_).

## Figures and Tables

**Figure 1 mps-05-00036-f001:**
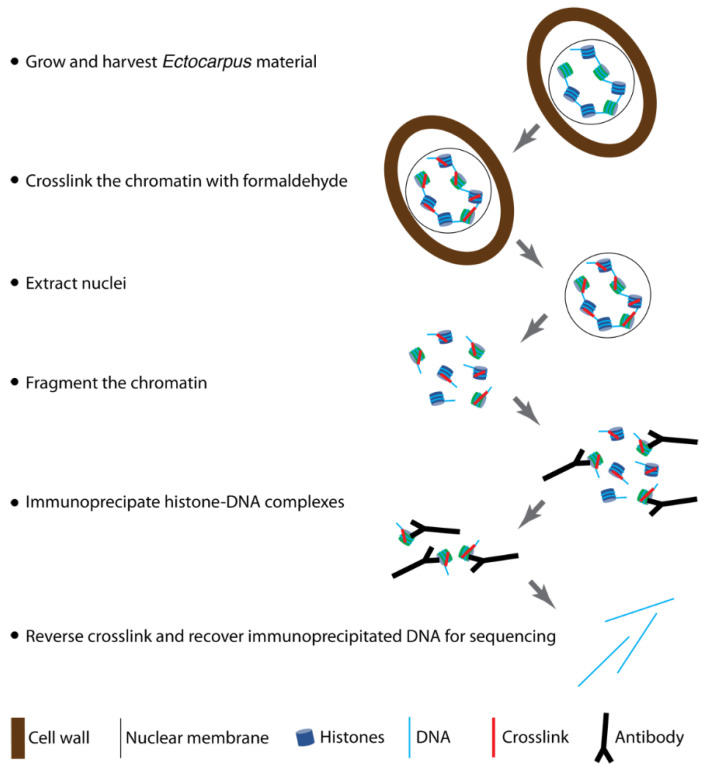
Overview of the ChIP protocol. After harvesting, the *Ectocarpus* material is treated with formaldehyde to crosslink the protein and DNA components of the chromatin. Chromatin is then extracted by isolating and subsequently lysing nuclei. After fragmentation of the isolated chromatin, DNA fragments crosslinked to histone proteins bearing a post-translational modification of interest are isolated by immunoprecipitating with an antibody raised against the histone PTM. Following reverse crosslinking, the isolated DNA can be recovered and sequenced.

**Figure 2 mps-05-00036-f002:**
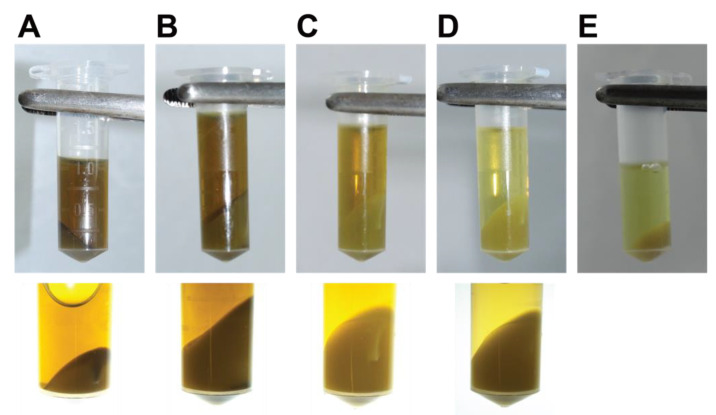
Aspect of the pelleted nuclei and chromatin preparations at different steps of the extraction procedure. (**A**) Centrifuged extract after Tenbroeck potter extraction and Miracloth filtering (step 13). (**B**) Centrifuged extract after the first wash with nuclei isolation buffer (step 14). (**C**) Centrifuged extract after the second wash with nuclei isolation buffer (step 14). (**D**) Centrifuged extract after washing with nuclei isolation buffer without Triton X-100 (step 15). (**E**) Centrifugation step after sonication to remove debris (step 22). The top panels show the whole tube and the bottom panels the aspect of the pelleted nuclei extracts.

**Figure 3 mps-05-00036-f003:**
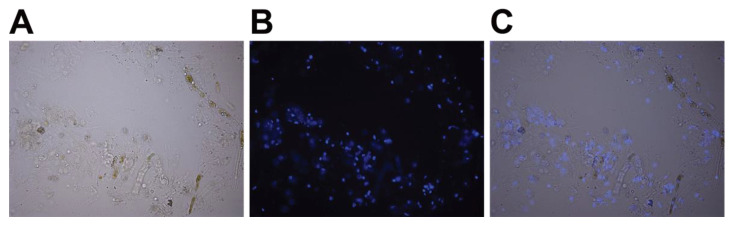
DAPI staining of extracted *Ectocarpus* nuclei. (**A**) Light microscopy image after disruption of thallus filaments by grinding in a Tenbroeck potter (step 16). (**B**) Fluorescence microscopy image of DAPI staining (blue colour). Extracted nuclei appear as bright spots. (**C**) Merge of the two images.

**Figure 4 mps-05-00036-f004:**
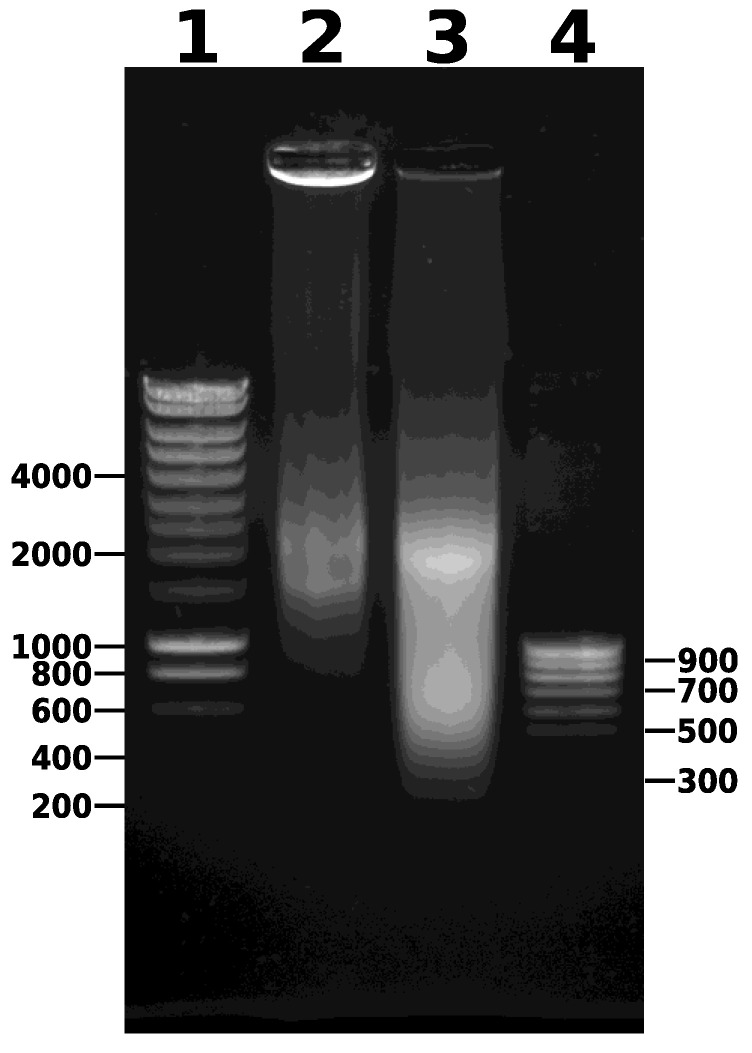
Agarose gel electrophoresis of unsonicated and sonicated chromatin. DNA size markers are indicated in bp, but note that the analysis only provides a rough indication of DNA fragment size because the chromatin is crosslinked. (**1**), 2.5 µL SmartLadder DNA size marker; 5 µL of extracted chromatin before sonication (step 19); (**2**), 5 µL of extracted chromatin before sonication (step 19); (**3**), 5 µL of extracted chromatin after sonication (step 22); (**4**), 2.5 µL SmartLadder SF.

**Figure 5 mps-05-00036-f005:**
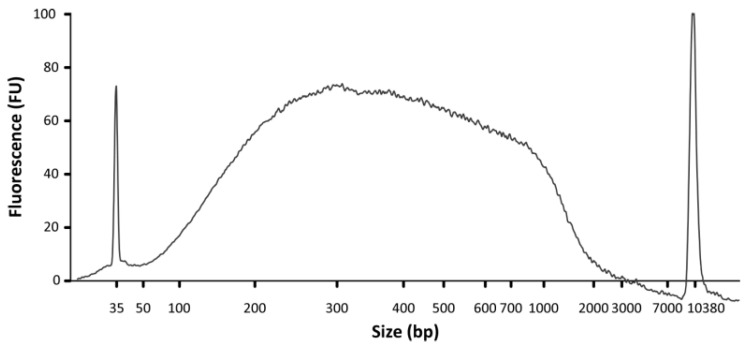
Example of a DNA electrophoregram profile obtained after chromatin immunoprecipitation. ChIP was performed using an anti-histone H3 (D2B12) XP^®^ rabbit monoclonal antibody (Cell Signal Technology). The fragmented DNA has a size range of 0.1 kbp to 1 kbp.

**Table 1 mps-05-00036-t001:** Approximate expected yields of DNA in immunoprecipitated chromatin for antibodies that recognize different histone PTMs in *Ectocarpus*.

Histone PTM or Control Sample	Antibody	DNA Yield/Fixed Material (ng/mg)
Input	n/a	10–25
Normal Rabbit IgG	(polyclonal) CST (ref:#2729)	0.02–0.05
Histone H3	(D2B12) CST (ref:#4620)	2–5
H3K4me3	(C42D8) CST (ref:#9751)	0.1–0.3
H3K9ac	(C5B11) CST (ref:#9649)	0.2–0.8
H4K20me3	(D84D2) CST (ref:#5737)	0.08–0.20
H3K27ac	(polyclonal) Millipore (ref:07-360)	0.05–0.30
H3K36me3	(polyclonal) Abcam (ref:ab9050)	0.03–0.20
H3K79me2	(D15E8) CST (ref:#5427)	0.20–0.40

## Data Availability

Not applicable.

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
