# Peer review of "An Efficient Chromatin Immunoprecipitation Protocol for the Analysis of Histone Modification Distributions in the Brown Alga Ectocarpus"

_mps, 2022, doi:10.3390/mps5030036_

Round 1
Reviewer 1 Report
Line 141 - "An agarose gel electrophoresis system" should be removed from "Materials and reagents";
Line 145 - ChIP grade antibody
Line 194, 197 - Centrifuge speed shoud be indicated in g instead rpm;
Line 350 - DEPC water should be mentioned in "Materials and reagents";
Line 354 - Bioanalyser, Qubit Fluorometer, High Sensitivity DNA Chip and dsDNA HS Assay kit should be mentioned in "Materials and reagents" and "Equipment".
Reviewer 2 Report
Dear authors,
I would like to acknowledge and congratulate you for such work, especially because I know how difficult it can be to deal with ChIP development.
Since I've already read your excellent "Histone Modifications during the Life Cycle of the Brown Alga Ectocarpus" article from the Genome Biology journal (2021), it's obvious to understand that this protocol can give expected results.
Your protocol is highly similar to some I've already read and practise, however I see you have made some adjustment, at least, in comparison with the protocols I know, with includes only animalia kingdom and I have clearly no knowledge on stramenopile ones, and I imagine that is one of the reasons.
I'm only wondering about the reason to use a sonication method with the Covaris, and not using Micrococcal nuclease (MNase), which give brighter/cleaner results (at least on invertebrates) - but I now also this step can be really tricky.. many be the MNase step didn't give expected results? Didn't work on stramenopiles?
By the way, this is my only question to this great protocol, and since this not really influence your protocol (it's more by curiosity, because I know that going through the sonication works anyway), for me there is no reason to don't accept directly your protocol.
The protocol is clear, steps are well presented, there is even a Figure that's describe what actually happen, for me English is clearly correct but I don't feel qualified to juge English language.
Reviewer 3 Report
The manuscript described a chromatin immunoprecipitation procedure for the brown algae Ectocarpus. The procedure is indicated as producing extraction of satisfactory amounts of chromatin and gives reproducible results, presenting them as yields in terms of microgram of extracted DNA and the size range of the fragmented DNA in the final samples.
I found this work very interesting and sounding for this research area, offering to the community clues to set up a powerful tool detecting in vivo interactions between a DNA-associated protein and genomic DNA in this particular and also other species. For this latter reason, my comments to the authors will address points that can be very useful to propose a stronger procedure.
Comments and corrections:
Line 174: change to “Petri”
Section 3.1.
For some organisms it is important the use of natural v/s artificial sea water to control the different components of the medium such as silica. Extraction of chromatin from some species can be difficult because of the silica based rigid cell wall which can interfere with chromatin extraction because it binds DNA. Authors should point out some comment about this setting (different use of silicon) to start the procedure; I’m thinking in general considerations that make possible to expand the use of the current procedure to other organisms.
Section 3.2. Excessive cross linking might reduce antigen accessibility and sonication efficiency. Authors should emphasize the relevance of the time used (under their experience) in this step and the effects of extending cross linking.
Fixed pellet can be stored at −80°C or do this need to be fresh for the protocol?
Section 3.4.
Is there any indication about sonication time? I think authors should comment their experience of the effect of trying different times and/or cycles in the species. Ideally, different times should be presented (or mentioned)
Section 3.5.
Materials for immunoprecipitation are critical in terms of number of cells in a given volume, and sample dilution can effectively decrease background noise. To avoid saturation of antibody in ChIP assays, calibration curves should be built before precipitation with the antibody to determine the optimal amount of antibody to use. I think this should be warned properly in this point of the protocol, or better, it should be presented for the current procedure.
Section 4. Have authors assayed some DNA quality check procedure in order to indicate reproducibility of the technique between samples or batches? For instance, using qPCR and the Ct value (cycles to cross the detection threshold) on duplicates can be an optimal way to visualize variability in the technique.
Section 5.1.
Is there any critical consideration for buffer and solution preparations such as a particular store temperature? I saw just some of them having indications.
